# DualLabel: Secondary Labels for Challenging Image Annotation

Chia-Ming Chang*
The University of Tokyo

Yi He**
JAIST

Xi Yang***
Jilin University

Haoran Xie†
JAIST

Takeo Igarashi‡
The University of Tokyo

## ABSTRACT

Non-expert annotators must select an appropriate label for an image when the annotation task is difficult. Then, it might be easier for an annotator to choose multiple "likely" labels than to select a single label. Multiple labels might be more informative in the training of a classifier because multiple labels can have the correct one, even when a single label is incorrect. We present *DualLabel*, an annotation tool that allows annotators to assign secondary labels to an image to simplify the annotation process and improve the classification accuracy of a trained model. A user study compared the proposed dual-label and traditional single-label approaches for an image annotation task. The results show that our dual-label approach reduces task completion time and improves classifier accuracy trained with the given labels.

**Keywords**: Annotation Tool, Challenging Image Annotation, Non-Expert Annotator, Secondary Label, Machine Learning.

**Index Terms**: • Human-centered computing~Human computer interaction (HCI)~Interactive systems and tools

## 1 INTRODUCTION

Accuracy is expected in annotations. Classifiers were trained and evaluated based on annotations, assuming they represent the ground truth. Humans provide imperfect annotations. This is especially true with limited budgets, and one cannot recruit a sufficient number of domain experts. One must use a limited number of annotators without sufficient domain knowledge (usually via crowdsourcing) [2, 10, 37] and train a classifier based on inaccurate training data. Multi-class labeling is particularly challenging for non-expert annotators where an annotator is required to select a single label among multiple "likely" options for a target image. Figure 1 shows an example of the difficulty faced by a non-expert annotator in a multiclass image classification task (i.e., a non-expert worker may not know the correct label, but they may know either label A or label B). Then, it is desirable to have a method to reduce the burden of annotations (making it easier for non-expert annotators to perform the task) while maintaining (or even improving) the accuracy of the classifiers trained with the resulting annotations without increasing the annotation cost.

We present a novel annotation tool, *DualLabel*, to solve this problem. The basic concept is that it is simple. When it is difficult for an annotator to select an appropriate label for a target image, we allow the annotator a second choice. We expect this to reduce

*e-mail: chiaming@ui.is.s.u-tokyo.ac.jp
**e-mail: s2010035@jaist.ac.jp
***e-mail: yangxi21@jlu.edu.cn
†e-mail: xie@jaist.ac.jp
‡e-mail: takeo@acm.org

the psychological burden of making a final decision and thus expedite the annotation process. This process produces more annotations for a set of images than the traditional single-label method; therefore, we expect that this method would improve the accuracy of a classifier trained with similar number of images.

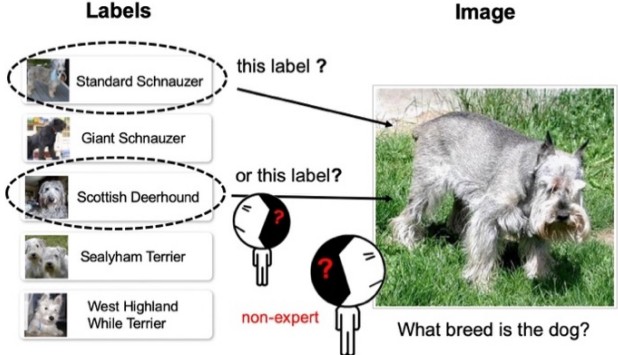

Figure 1: Problem of multi-class labeling for non-expert annotators.

We conducted a user study to compare the proposed dual-label approach with a traditional single-label approach for multi-class image annotation. We measured the time required for annotation and collected the subjective feedback. We also trained classifiers using the annotation results and measured their accuracy. The results showed that the dual-label approach requires less time to complete the image annotation task than the single-label approach in a challenging image annotation task. The classifier accuracy trained with dual labels was better than that of the classifier trained with single labels. Our questionnaire results showed that participants felt that the dual-label approach was more straightforward, helpful, and efficient than the single-label approach when they were not confident in their label selection during annotation. Additionally, the dual-label approach increased the annotators' confidence in challenging image annotation. This study makes the following contributions.

- *DualLabel*, a novel annotation tool that allows users to assign a secondary label to a difficult multiclass image classification task.

- A user study comparing *DualLabel* to a traditional single-label approach demonstrated the benefits of the dual-label approach in challenging image annotation tasks.

- A machine learning experiment training classifier using two sets of annotations showed that the dual-label approach could improve the accuracy of classifiers.

## 2 RELATED WORK

### 2.1 Crowdsourcing Data Annotation

Crowdsourcing is a popular method for conducting data-annotation tasks. Data quality in crowdsourcing often contains numerous errors [3, 4, 5]. Crowdsourcing data annotation tasks are often designed to be relatively simple and easy to increase the data quality. For example, various types of binary tasks (i.e., users selecting one option from two) are used. In a binary task, crowd workers are requested to answer a simple yes or no question, such as "does this image contain a dog?" ImageNet [1] is a large-scale image dataset manually verified by humans via binary judgment. Pairwise HITS [19] is an annotation workflow that allows annotators to compare a pair of labeled data and select the better one. A binary labeling task is easy when the data are clear or the task is simple (e.g., selecting a dog or cat label). However, it becomes difficult for annotators when the data are difficult or when the task is complex (e.g., selecting a dog breed label for a dog image). Then, the data quality may be unstable and low, which causes problems in the machine learning process [22].

To avoid errors during annotation, an "unsure" function is conventionally provided during annotation, which allows annotators to give up (skip) a labeling task when it is difficult [6, 7]. For example, Audio Set [17] and Revolt [18] provided uncertain features in their annotation tasks. This "unsure" option can avoid errors made by the annotators when they are not sure about the data. In addition, "user confidence" is another piece of information that can be collected during annotation to measure label quality. For example, annotators indicate their confidence scores on the label they select for an image. Oyama et al. [21] proposed a labeling approach that allows annotators to assign confidence scores to a binary labeling task. Jinhua et al. [21] proposed a method that shares the same concept in a data annotation task. Here, annotators were asked to judge their answers by giving confidence via a sliding bar.

These approaches primarily focus on single-label annotation tasks. However, this may be insufficient for challenging data-annotation tasks. For example, it can be very difficult for annotators to select an appropriate label from two possible labels when they are not confident in the labels. We believe that our dual-label approach can provide new opportunities to address this issue, different from other existing approaches.

### 2.2 Utilizing Different Data in Machine Learning

Uncertain functions have been widely used in various data-annotation tasks. Several studies have proven that uncertain data can help machine learning results. Takeoka et al. [8] indicated that the performance of unsure loss is better than that of conventional models (no unsure labels) because unsure labels tend to be located close to the ground-truth decision boundary. Zhong et al. [6] indicated that providing annotators an uncertain option would significantly benefit active learning from crowds (ALC). Cui et al. [51] proposed an uncertainty pairwise comparison oracle to aid interactive labeling by comparing the uncertainty of two unlabelled data points. In addition, Wu et al. [9] proposed a unified end-to-end learning framework by using "unsure data" in medical image analysis. This framework shows the benefits of learning with uncertain data, and the validity of their models is demonstrated in the prediction of Alzheimer's Disease and lung nodules.

Alternatively, Ishida et al. [11] introduced a learning approach learning from complementary labels, for optimizing algorithms. A complimentary label specifies the class to which the pattern does not belong. This can reduce the human workload in manual data annotation because selecting an incorrect label is easier than selecting a correct label when data is difficult. Yu et al. [12]

shared this concept and proposed a framework for learning with biased complementary labels. Moreover, partial label learning is an approach for training a set of possible labeled data, where each instance is tagged with more than one label, only one of which is correct [13, 23, 24, 52, 53]. These studies focused on partial multi-labels for machine learning, whereas our study focused on the annotation process. Cui and Sato [50] introduced a method for learning from triplet comparison data using only passively obtained triplet comparison data. Whitehil et al. [14] introduced GLAD, a learning model for optimizing the integration of labels from annotators of unknown expertise. This model refers to the concept "whose vote should count more" that can recover the true data labels more accurately than the Majority Vote heuristic.

Various systems/algorithms use different types of collected data to improve machine learning results. Developing a better system/algorithm to enhance machine learning accuracy is a popular (common) approach that most studies have focused on. However, there is also another approach which is to provide better data (i.e., correct labels). This study focused on the latter and aimed to expedite the data annotation process from a human perspective.

### 2.3 Expediting Data Annotation Process

The data annotation process can be discussed in terms of two aspects: (a) data quality and labeling efficiency (i.e., objective data analysis) and (b) the annotator's perception during annotation (i.e., subjective data analysis). Many studies have focused on the former aspect and have proposed efficient and supportive approaches to expedite the data annotation process (i.e., decrease labeling time and increase label quality). For example, a hierarchical task assignment was proposed to expedite manual image annotation [28]. Moreover, many semi-automatic annotation systems have been proposed to assist manual image labeling using collaborative filtering and computer vision techniques [29, 30, 31]. Interactive concept learning guides annotators in assigning labels to the most informative images for classifiers [32, 33, 34].

Some studies have focused on annotators' perceptions. For example, Ahn et al. [35] proposed an image annotation tool combined with a computer game. Chang et al. [36] designed a spatial layout labeling interface that can increase the confidence of nonexpert annotators in an image annotation task. In addition to non-expert annotation, Schaekermann [57] explored the understanding of expert disagreements in medical image annotation. Some studies have indicated that the human aspect is an important part of crowdsourcing annotation tasks. For example, Zhuang and Gadiraju [38] analyzed crowd workers' mood, performance, and engagement during annotation; LaPlante et al. [39] investigated the trust issue between crowd workers and task requesters; and Durward et al. [40] identified ethical issues in crowdsourcing, especially with a focus on the crowd workers' perspective. Chang et al. [56] investigated the effects of quick and careful labeling styles on an image annotation task.

These studies have shown that human aspects play an essential role in manual data annotation. However, no detailed research has explored the relationship between the data quality/labeling efficiency (i.e., objective data analysis) and the annotator's perceived usability (i.e., subjective data analysis). This can be a significant point in expediting the data annotation process investigated in this study.

## 3 DUAL LABEL

We propose *DualLabel*, a dual-label image annotation tool that allows annotators to assign a high-confident and low-confident (secondary) label to an image when making a label decision is challenging during annotation (i.e., not confident with the target

image and labels). Figure 2 shows a screenshot of the dual-label annotation interface using dog-breed labeling. The lower part of the left-side interface shows the labels to be assigned to the corresponding representative images, and the upper part presents a magnified view of the activated label. The user activates the label by clicking on one of the labels below. The upper part of the right-side interface represents the target image to be labelled. The lower part of the right-side interface is the selected label(s) for the target image, which includes a high-confidence label box (left) and a low-confidence label box (right). The annotator drags a label on the bottom left of the label box on the bottom right to assign a label. A high-confidence label is required, but a low-confidence label is optional (annotators are allowed to choose one label they are confident of). Annotators were allowed to swap the high-confidence and low-confidence labels by clicking the arrow icon between the two boxes.

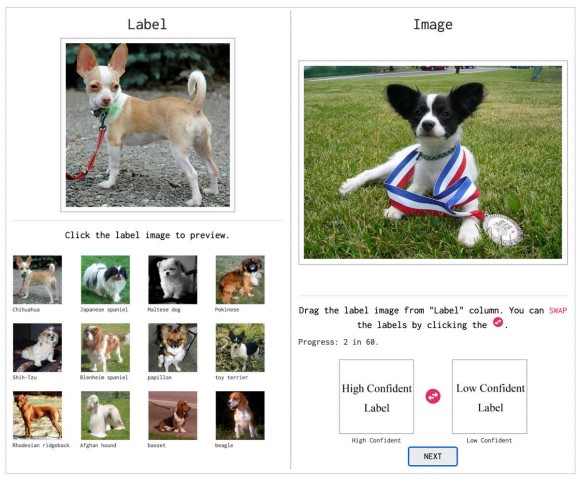

Figure 2: *DualLabel* user interface.

Figure 3 shows an example of a usage scenario in a challenging image annotation task using a dual-label image annotation tool. Here, the annotator is not confident about the target image and labels (i.e., it is difficult to make a label decision). Then, the annotator is unsure which dog breed (label) is the most appropriate label for the target dog image, but they think that the answer is probably either "*Chih-Tzu*" or "*Maltese dog*". So, the annotator selects "*Chih-Tzu*" as a primary label (high-confident label) and "*Maltese dog*" as a secondary label (low-confident label) for the image.

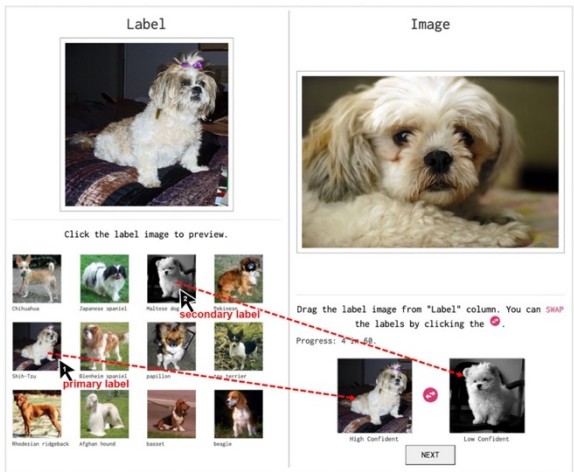

Figure 3: An example of a usage scenario.

This dual-label annotation tool helps annotators make a label decision more easily and quickly when they face difficulties making an appropriate label decision during annotation. We assume that in most cases, the two selected labels may contain one "correct" label and one "likely" label. We expect that this "likely" label is a valuable resource that improves the classification accuracy of a trained model.

## 4 USER STUDY

A user study compared a traditional single-label annotation approach with our proposed dual-label annotation approach for a manual image annotation task. Our hypothesis is that the dual-label approach can expedite the annotation process in challenging image classification, mainly when the annotators are not confident in label assignment.

### 4.1 Image Annotation System

Two online image annotation systems single -and dual-label) were developed for the user study. We used HTML and JavaScript for user interface and Node development. js on the server-side to collect and process the data. The system was deployed on an Ubuntu system with 2GB of RAM provided as a server from the Sakura VPS. The image-annotation interfaces are shown in Figure 4. The only difference between the two interfaces is that the single-label interface has only one label box (participants select only one label for an image), whereas the dual-label interface has two label boxes (participants can select a low-confidence label in addition to a high-confidence label for an image).

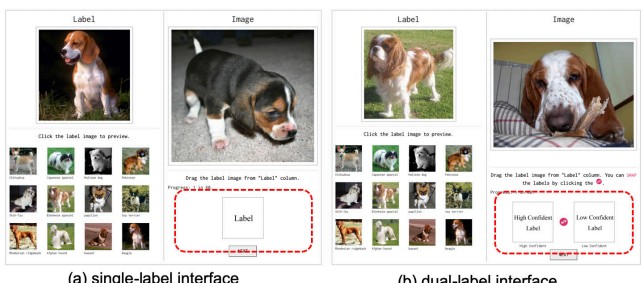

(a) single-label interface          (b) dual-label interface

Figure 4: Screenshots of single-label and dual-label image annotation interfaces.

### 4.2 Participants

Twenty-four participants (16 men and 8 women, aged 18 to 59 years) were recruited using Amazon Mechanical Turk (MTurk) [25]. All participants were MTurk Master Workers and had a 98% HIT approval rate [26]. Eighteen of the 24 participants (75%) have (or had) dogs as pets. However, none of the participants had professional knowledge of the dog breeds. We paid a slightly higher compensation ($15.75/h) to the participants than the average wage in Amazon MTurk ($11.58/h) [27].

### 4.3 Dataset

We used a dog image dataset from ImageNet (ILSVRC 2012), which contained 120 dog labels (breeds) [15]. We manually created two datasets (Datasets A and B) with different difficulty levels for the image annotation tasks. The difficulty levels of selected dog labels (breeds) are based on Stanford dog image classification [16]. The Stanford Dogs Dataset also used 120 dog labels from ImageNet. Their machine learning experiment followed a similar training and testing methodology as Caltech-101 [49]. Dataset A includes the most difficult 12 dog breeds (among the 120 dog breeds), and Dataset B included the easiest 12 dog breeds. A difficult dog breed means that the classification

accuracy is low for the dog breed [16]. Tables 1 and 2 show the 12 dog breeds in Datasets A (difficult) and B (easy). For each dog breed (label), five images were randomly selected from 1,300 images, from ImageNet (ILSVRC 2012) [15] for the image annotation task in the user study.

Table 1. Dataset A: 12 difficult dog breeds.

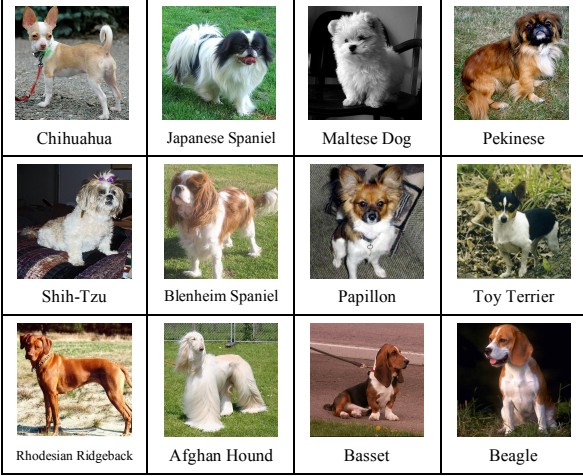

Table 2. Dataset B: 12 easy dog breeds.

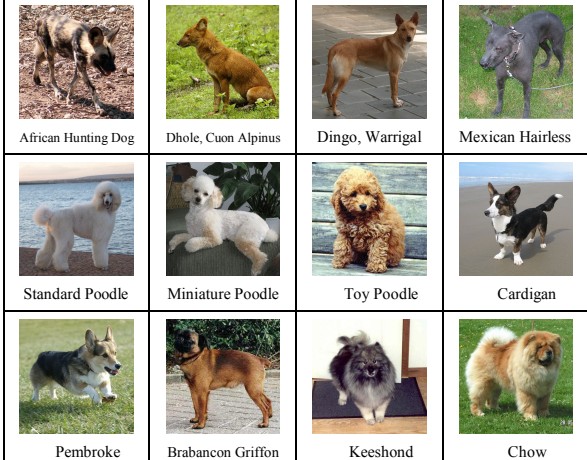

### 4.4 Task and Condition

The image annotation task for each participant involved labeling 120 dog images (60 = 12 × 5 images for each dataset). The annotation task requested participants to select an appropriate dog breed (label) for each dog image from a 12-dog breed list (12 labels). The within-subjects method was used, where 24 participants were asked to complete two annotation tasks via the single-label and dual-label interfaces (different datasets were used for the tasks conducted by different interfaces).

**Single-label Condition.** This is the baseline. Figure 4 (a) shows a screenshot of the single-label annotation interface. The instructions for the single-label annotation task were as follows.

*"If you are confident with the target image and the label you intend to select, assign it to the image directly. However, if you are not confident with the target image and the label, please consider carefully and try to assign the most appropriate one."*

**Dual-label Condition.** This was the proposed approach. The dual-label annotation interface is shown in Fig. 4 (b). The instructions for the dual-label annotation task were as follows.

*"If you are confident with the target image and the label you intend to select, assign it to the image as the 'high-confident label'. However, if you are not confident with the target image and the label, assign the label you are more confident with to the high-confident label box, and the label you are less confident with to the low-confident label box."*

We balanced the order of the conditions and datasets among the 24 participants to ensure a balance between the two conditions and the two datasets. Table 3 shows the distribution of the conditions and datasets.

Table 3. Distribution of the training data.

| Participants | 1st Task | 2nd Task |
|---|---|---|
| P01 - P06 | Single-Label + Dataset A | Dual-Label + Dataset B |
| P07 - P12 | Single-Label + Dataset B | Dual-Label + Dataset A |
| P13 - P18 | Dual-Label + Dataset A | Single-Label + Dataset B |
| P19 - P24 | Dual-Label + Dataset B | Single-Label + Dataset A |

### 4.5 Procedure

A 3-minutes video instruction was presented at the beginning to the participants to explain the details of the user evaluation process. This includes a step-by-step demonstration (step by step) of how to use the annotation interfaces to complete the given image annotation tasks. After completing the two annotation tasks, the participants were asked to complete a questionnaire about the annotation tasks.

### 4.6 Measurement

Our annotation system recorded and measured the time and accuracy of the image annotation tasks completed by the participants. The system also recorded the time spent by participants on each image label. The system recorded the number of secondary labels collected via the dual-label approach in cases (images) when the participants were not confident with a single label selection. The questionnaire had three Likert-scale questions and one open-ended question regarding the participants' perception and preference of the single-label and dual-label annotation approaches.

### 5 MACHINE LEARNING EXPERIMENT

In addition to evaluating the manual image annotation process, we conducted a small-scale machine learning experiment to compare the accuracy of classifiers trained with the labeled data created using the single-and dual-label approaches. The hypothesis is that the labeled data collected using the dual-label approach achieve better classification accuracy than the data collected via the traditional single-label approach.

**Training Data.** We used data collected from the two annotation approaches to train the classifiers. A total of 2,880 labeled images were collected from the user study. Table 4 presents the distribution of training data. A total of 720 labelled images (60 labelled images for each dog breed) were collected for each condition. The number of images was similar (720 images); however, the dual-label task contained more labels for training. The collected training data contained errors made by the participants (i.e., the accuracy rate of the training data was not 100%).

Table 4. Distribution of the training data.

| Tasks | Dataset | Amount |
|---|---|---|
| Single-label | Dataset A (difficult) | 60x12=720 |
| Single-label | Dataset B (easy) | 60x12=720 |
| Dual-label | Dataset A (difficult) | 60x12=720 |
| Dual-label | Dataset B (easy) | 60x12=720 |

**Testing Data.** The test data used in the machine learning experiment were 2,880 images (60 × 12 =720 images in each condition). These images were not included in the training dataset.

**Classification Algorithm.** We used the unpretrained AlexNet [46] implemented by PyTorch [47]. We trained four classifiers, each trained using 720 images, as listed in Table 4. The accuracy of each classifier was measured by using 720 test images (60 images from each class). We reshaped the input images to a size of 224×224 and trained the model with a learning rate of 0.01 and a weight decay of 0.0005 as parameters. We used the stochastic gradient descent (SGD) optimizer, passed the model's output to a sigmoid layer, and then obtained the binary cross-entropy loss. The number of epochs of the model training was 200. For the data collected via the dual-label approach, we encoded all the categories into 0-1 vectors while assigning a weight of 1 to the high-confidence label and 0.5, to the low-confidence label. The sum of the vectors was normalized to one. The output is consistent with that of the single-label approach; both are single-result outputs. We then compared the accuracy of the labeled data obtained using the two annotation approaches on the test data.

## 6 RESULTS

### 6.1 Task Completion Time

**Overall Task Completion Time.** Figure 5 shows that the participants spent an average of 10 min 6 s and 10 min 37 s labeling the 60 images using the single-label and dual-label approaches. The results of the paired t-test on task completion time showed that the difference was not significant ($p > 0.05$) between the single-label and dual-label approaches. This indicates that the dual-label task does not require additional time to complete the image annotation task.

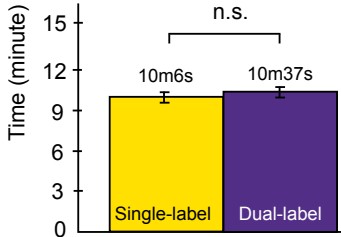

Figure 5: Overall task completion time. Single: mean = 10.06; SD = 3.76; Dual: mean = 10.37; SD = 3.90.

**Task Completion Time in Easy and Difficult Datasets.** Figure 6 shows the task completion times for the easy and challenging datasets. The accessible dataset shows that the participants spent an average of 7 min 49 s and 11 min 39 s labeling the 60 images via the single-label and dual-label approaches, respectively. Although the time appears different between the two approaches, the result of the paired t-test showed that the difference was not significant ($p > 0.05$). This was because the dual-label task's standard deviation (SD) was high. In the difficult dataset, the participants spent an average of 12 min 25 s and 9 min 36 s using the single-label and dual-label approaches, respectively. The results of the paired t-test showed that the

difference was significant ($p < 0.05$) between the two approaches. This indicates that the dual-label approach is more efficient (i.e., it requires less time to complete the task) than the single-label approach when the target images are difficult.

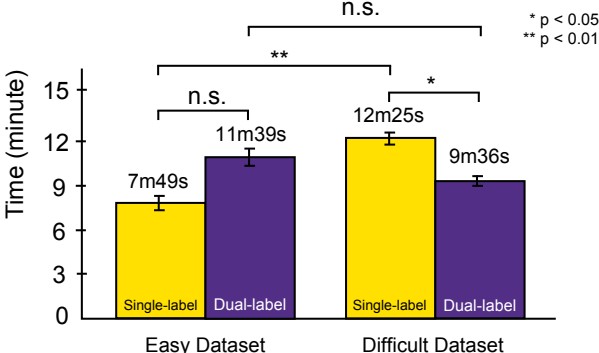

Figure 6: Task completion time in the easy and difficult datasets. Easy Dataset. Single: mean = 7.49; SD = 3.24; Dual: mean = 11.39; SD = 4.54. Difficult Dataset. Single: mean = 12.25; SD = 2.70; Dual: mean = 9.36; SD = 2.76.

In addition, the results show that, in the single-label task, the participants spent significantly ($p < 0.01$) a more extended time completing the challenging task (12 m25s) and the easy task (7m49s). However, the difference between the easy (11m39s) and complex (9m36s) tasks in the dual-label task was not significant ($p > 0.05$). This indicates that the dual-label approach is robust against task difficulty in terms of task completion time.

### 6.2 Annotation Accuracy

Figure 7 displays the accuracy of the annotations provided by the participants using the single- and dual-label approaches. The results show that the annotation accuracy was 75% for the single-label task, while the annotation accuracy was 71.12% for the high-confident label and 12% for the low-confident label in the dual-label task. The results of the paired t-test showed that the difference was not significant ($p > 0.05$) between the labels in the single-label task and the high-confidence labels in the dual-label task. This indicates that the dual-label approach cannot significantly improve label quality; however, it can collect more data (i.e., low-confidence labels) during annotation. Additionally, the results showed that the accuracy of the low-confidence label in the dual-label task was only 12%. This indicates that the low-confidence labels are not very reliable.

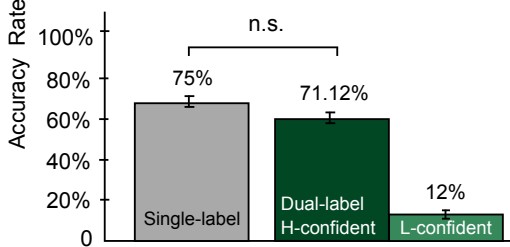

Figure 7: Accuracy rate. Single: mean = 75.00; SD = 9.65; Dual-label H-confident: mean = 71.12; SD = 8.98 and L-confident: mean = 12.00; SD = 6.26.

Figure 8 shows the annotation accuracy of the annotation tasks for the easy and difficult datasets. The easy dataset shows that the accuracy rate was 78.42% for the single-label task, while the accuracy rate was 70.50% for the high-confident label and

13.39% for the low-confident label in the dual-label task. In the difficult dataset, the results show that the accuracy rate was 71.58% for the single-label task, while the accuracy rate was 71.75% for the high-confident label and 10.08% for the low-confident label in the dual-label task. The results of the paired t-test showed that the difference was not significant (p > 0.05) between the annotation accuracy in the single-label task and the dual-label task (high-confidence label) for both the easy and difficult datasets.

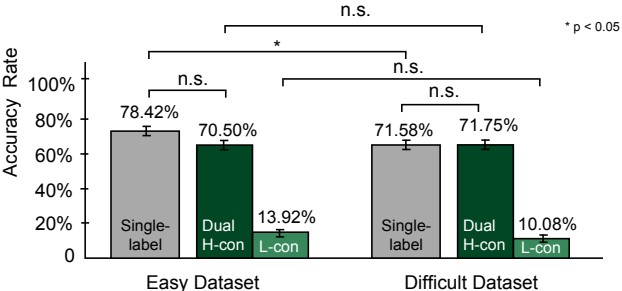

Figure 8: Accuracy rates in the easy and difficult datasets. Easy Dataset. Single: mean = 78.42; SD = 6.86; Dual_H-con: mean = 70.50; SD = 8.61 and L-con: mean = 13.92; SD = 7.61. Difficult Dataset. Single: mean = 71.58; SD = 11.07; Dual_H-con: mean = 71.75; SD = 9.69 and L-con: mean = 10.08; SD = 4.01.

The results showed that the difference was significant (p < 0.05) between the two datasets in the single-label task. However, the difference between the two datasets in the dual-label task was not significant (p > 0.05). This indicates that the data difficulty affects only the annotation accuracy in the single-label task. This does not affect the label quality in the dual-label task.

### 6.3 Number of Assigned Labels

Table 5 shows the number of assigned labels collected via single - and dual-label annotation tasks in the user study. In the single-label annotation task, the total number of labels assigned to images was 1440 labels (720 for Dataset A and 720 for Dataset B). In the dual-label annotation task, the total number of labels assigned to images was 2397 (66.45% more than the single-label task), which includes 1440 high-confidence labels (720 for Dataset A and 720 for Dataset B), and 957 low-confidence labels (481 for Dataset A and 476 for Dataset B).

Table 5. Distribution of the training data.

| Tasks | Datasets | Number of Assigned Labels |
|---|---|---|
| Single-label | Dataset A | 720 |
| | Dataset B | 720 |
| Dual-label | Dataset A | 720 (h-con) and 481 (l-con) |
| | Dataset B | 720 (h-con) and 476 (l-con) |

(h-con = high-confident label; l-con = low-confident label)

The results show that the dual-label annotation task can collect 66.8% (Dataset A) and 66.11% (Dataset B) more labeled images (i.e., low-confidence labeled images) than the single-label annotation task. Originally, expected the dual-label approach would collect more data (i.e., low-confidence labeled images) in the difficult dataset than in the easy dataset because the annotators might find it difficult to make a single label decision when the images are difficult. However, the results showed almost no difference between the easy (481 images) and difficult (476 images) datasets. This indicates that the data difficulty does not affect the amount of data collected in the dual-label annotation task.

### 6.4 Learning Effect

Figure 9 shows the average time for the annotation process for the first half (1-30 images) and the second half (31-60 images) via the single-label and dual-label approaches in the easy dataset. The results showed that the participants spent an average of 4 min 11 s and 3 min 38 s to complete the first half and second half, respectively, using the single-label approach, whereas they spent an average of 5 min 21 s and 6 min 18 s, respectively. The paired t-test showed no significant difference (p > 0.05) between the first and second halves of the annotation process in both the single-label and dual-label annotation tasks. This indicates that there is no learning effect for easy images.

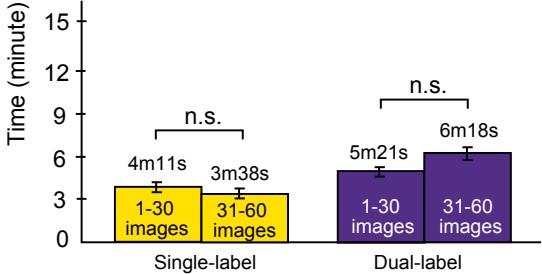

Figure 9: Average time for the annotation process in the easy dataset. Single (1-30 images): mean = 4.11; SD = 1.95; Single (31-60 images): mean = 3.38; SD = 2.12. Dual (1-30 images): mean = 5.21; SD = 3.43; Dual (31-60 images): mean = 6.18; SD = 4.13.

Figure 10 shows the average time for the annotation process for the first half (1-30 images) and the second half (31-60 images) via the single-label and dual-label approaches in the difficult dataset. The results showed that the participants spent an average of 6 min 3 s and 6 min 22 s to complete the first half and second half, respectively, using the single-label approach, whereas they spent an average of 5 min 39 s and 3 min 57 s, respectively. The results of the paired t-test showed no significant difference (p > 0.05) between the first and second halves of the annotation process in the single-label annotation task. However, a significant difference (p < 0.05) was observed in the dual-label annotation task. This indicates that the time per image decreased in the second half of the annotation process. This implies that annotators can learn and improve the efficiency of the annotation process using the dual-label approach when images are difficult.

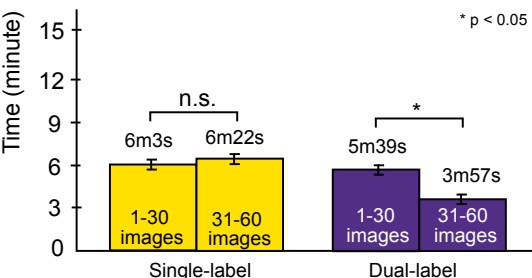

Figure 10: Average time for the annotation process in the difficult dataset. Single (1-30 images): mean = 6.03; SD = 2.52; Single (31-60 images): mean = 6.22; SD = 2.37. Dual (1-30 images): mean = 5.39; SD = 1.01; Dual (31-60 images): mean = 3.57; SD = 1.39.

### 6.5 Questionnaire

Figure 11 shows questionnaire results. Fig 11 (a) shows that the dual-label approach can reduce annotators' subjective feelings of difficulty when the images are difficult in the annotation task. Fig 11 (b) shows that the dual-label approach was more helpful than

the single-label approach when the images are difficult in the annotation task. Fig 11 (c) shows that the dual-label approach was more efficient than the single-label approach when the images were difficult in the annotation task.

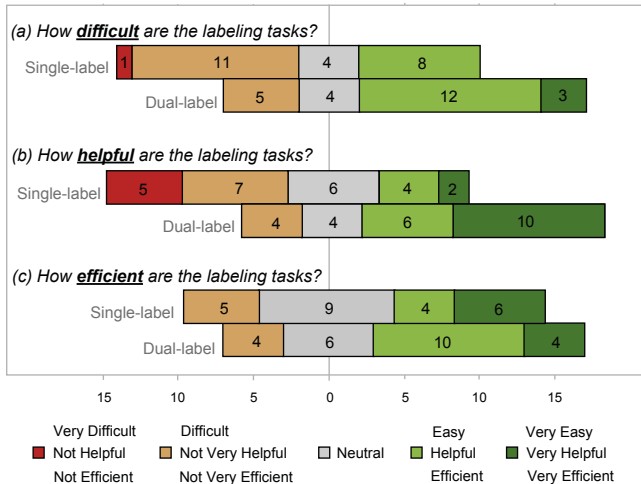

Figure 11: Questionnaire results.

The results also showed that 66.67% of the participants (n = 16) preferred the dual-label approach, while 33.33% preferred the single-label approach (n = 8). Participants who preferred the dual-label approach indicated that the dual-label annotation interface was supportive and helpful. For example, one participant indicated that "*Even if I got my answer wrong on the most confident, between the two of them I felt I was probably right on one of them*"; one participant indicated that "*having a more confident/less confident choice gave more leeway into picking the right breed, especially the ones that were hard to tell/could have been more than one*"; and another participant indicated that "*I felt like using two labels was insurance against being wrong so I felt less anxiety about my choices. I just felt more comfortable working on the task when I could choose two labels instead of one.*" In contrast, participants who preferred the single-label approach indicated that the single-label annotation interface was easy and simple. For example, one participant indicated that "*I feel like having two options made me second-guess my instincts a little bit*" one participant indicated that "*most of the dog images were clear for me, I only need the single-label interface,*" and another participant indicated that "*it's an easy mode for the annotation task*".

### 6.6 Results of Machine Learning Experiment

Figure 12 shows the accuracy of the classifiers trained with the labeled data collected under each annotation condition (each bar corresponds to a classifier trained with 720 images). The results show that the dual-label approach increases the classifier accuracies by 6.95% (from 78.19% to 84.03%) and 9.86% (from 71.53% to 81.94%) in the easy and difficult datasets, respectively. The paired t-test (for each dog breed, we compared the accuracy of single-label and dual-label as a pair) on classification accuracy showed that the difference was significant ($p < 0.01$) between the single-label and dual-label approaches in the difficult dataset, but not significant ($p > 0.05$) between the two approaches in the easy dataset. This indicates that the labeled data collected via the dual-label approach can improve the accuracy of classifiers in difficult datasets (i.e., challenging image annotation).

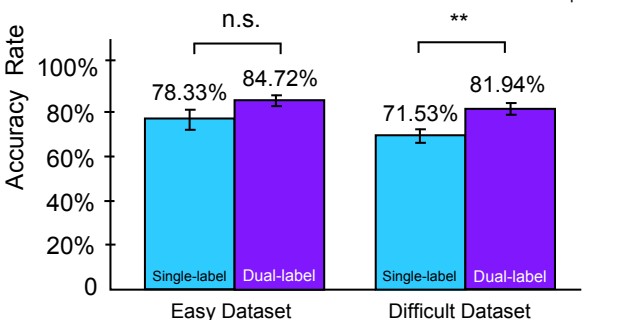

Figure 12: Accuracy rates of machine learning performance. Easy Dataset. Single: mean = 78.33; SD = 19.38; Dual: mean = 84.72; SD = 9.27. Difficult Dataset. Single: mean = 71.53; SD = 17.47; Dual: mean = 81.94; SD = 14.78.

## 7 DISCUSSION

### 7.1 Dual-label Approach Expedites Annotation Process in Challenging Image Classification

Manual image annotation is tedious and time-consuming and often relies on crowded workers. It is demanding for non-expert crowd workers (who have insufficient domain knowledge) to select the correct label when the task is difficult [3, 4, 5]. The dual-label approach allows annotators to select two labels for an image when making a single-label decision is difficult. Our original concern was that the selection of two labels for an image might take more time than just selecting one label, because annotators need to repeat the same operation twice (i.e., selecting a label for an image). However, the results show that the dual-label approach performed better than expected in the user study. This indicates that the participants spent significantly ($p < 0.05$) less time completing the image annotation task in the dual-label approach than in the single-label approach when the image annotation task was difficult. This implies that the dual-label approach expedites the manual image annotation process (i.e., reduces task completion time) in challenging image classification. Reducing image annotation time brings significant benefits to manual data annotation because we can reduce the cost of conducting a manual data annotation task (normally, labor-intensive, and costly).

### 7.2 Dual-label Approach Shows Learning Effect During Annotation in Challenging Image Annotation

The learning effect describes how people learn during a given process. This phenomenon analyzes the efficiency of activity or study. Learning effects have been studied and discussed in various areas [44, 45, 48]. It is also used in data annotation to analyze task performance (i.e., efficiency) during the labeling process [28]. We analyzed the learning effect during given image-annotation tasks in the user study. The results showed a significant learning effect ($p < 0.05$) during the annotation process in the dual-label annotation task when the images were difficult. The dual-label approach allows the annotators to "learn" during annotation and increases the annotation efficiency (i.e., reduces the task completion time in the second half of the image annotation task) in difficult image classification. The reason for this effect remains unclear. One possible explanation for this is that it helps annotators build domain knowledge during the process. It also allows annotators to learn how to use the user interface. We conjecture that the second one may not be because

we did not see a similar learning effect in the easy image-labeling task. These results have shown that the learning effect is an essential factor during annotation, and it has provided potential insights for the future development of annotation tools, such as designing a learnable annotation tool (e.g., learning domain knowledge during annotation).

## 7.3 Dual-label Approach Collects More Data and Improves Classifier Accuracy

In general, data provided better classification accuracy. Usually, more human effort and money are necessary for a manual data annotation task to collect more training data. The results showed that the dual-label approach could collect more data (labeled images) without significantly increasing the time (even requiring less time when the images are difficult). More specifically, the dual-label approach collected 481 (66.8%) more labeled images (i.e., low-confidence label) than the single-label approach when the annotation task was difficult and 476 (66.1%) more labeled images when the annotation task was easy. These additional data collected via the dual-label approach is the "low-confident" labeled images. The results of our machine learning experiment show that these 66.8% and 66.1% additional data improved the classification accuracy by 9.86% (difficult dataset) and 6.95% (easy dataset), respectively. This improvement was affected by the low-confidence labeled images collected via the dual-label approach. This result implies that low-confidence data is significant in the progress of classification accuracy.

The dual-label approach expedites the image annotation process and improves classification accuracy. We believe that this significantly benefits difficult image annotation and provides valuable insight for conducting a crowdsourcing annotation task. Note that the dual-label approach collects more labels without requiring additional annotation time and increases the number of annotations (label assignments) without increasing the number of images to be annotated. This is significantly more efficient than increasing the number of annotations by expanding the number of images to be annotated. An annotator spends a significant amount of time observing the target image to assign the first label, and dual label approach "reuses" the effort in assigning the second label.

## 7.4 Dual-label Approach Increases Perceived Usability and Worker's Confidence in Challenging Image Classification

The analysis of subjective feedback from the questionnaires shows that the dual-label approach can increase the perceived usability of crowd workers in challenging image annotations. Over half of the participants felt that the dual-label approach was easier (n = 15), more helpful (n = 16), and more efficient (n = 14) than the single-label approach. One participant commented, "*The dual-label interface is more preferable since it allows one to enter their top two choices, but doesn't mandate them to do so. This is both efficient as it doesn't require easy judgments to be more complicated than they are yet simultaneously allowing one to put their top two selections without spending an inordinate amount of time internally debating about which answer they should select*." This shows the benefits of the dual-label approach from a human perspective (e.g., annotators' perception). This is an important result, as several studies [38, 39, 40] have indicated that the human aspect is a critical issue when conducting crowdsourcing tasks. The dual-label approach increases the confidence of crowd workers during annotation. For example, one participant indicated, "*The dual label interface leads to much higher confidence when selecting an answer, especially for breeds that*

*are very similar in appearance*." Another participant indicated, "*I feel bad when I make mistakes. With the dual-label interface, I feel less likely to give a wrong answer. Also, I spent less time choosing a label because I felt more confident when I could pick two labels*." Confidence is a factor that affects the efficiency of manual data annotation in challenging image annotation, as it is an essential factor that improves people's motivation for learning and its efficiency [41, 42, 43].

## 8  LIMITATION AND FUTURE WORK

A limitation of this study is that the training data used in the machine learning experiment were small (60 images for each dog breed) and were only tested in a limited setting. The primary purpose of this study is not to pursue high accuracy rates of machine learning performance but instead to focus on the effects of the dual-label approach during annotation in challenging image classification. Our results show that the dual-label approach can collect more data, and the data can increase machine learning accuracy. We believe that the dual-label approach may have an even greater effect on large-scale data annotation tasks. This study only evaluated the dual-label approach with a type of data and task (dog image classification task). We believe that the dual-label approach can also be used to benefit various data types and annotation tasks.

In the future, we would like to conduct a large-scale user study and test more machine learning conditions (e.g., soft labeling [54] and self-knowledge distillation [55]). There are many possible variations in using collected data with the dual-label approach. For example, over two labels (three or four) label an image, or force two labels to an image. They are applying a dual-label to binary classification (no label to an image might differ from two labels to an image). We investigate the real reasons for the learning effect (improving annotation efficiency). One possible direction is to observe the annotators' behavior during the entire annotation process. For example, how do workers spend their time during an annotation task (i.e., what exactly do they do?). Another possible direction is to explore self-learning during annotation and design a learnable annotation interface to help non-expert annotators build domain knowledge during annotation.

## 9  CONCLUSION

This study presented *DualLabel*, a new manual image annotation tool for expediting the manual data annotation process and improving the classification accuracy of the trained model in challenging image annotation. This tool allows annotators to assign a high-confidence and low-confidence label to an image when the annotators find it challenging to make a label decision. The user study compared the proposed dual-label approach to a traditional single-label annotation task for a manual multi-class image annotation task. The results showed that our dual-label approach could collect more data with a shorter task completion time when the images were difficult. The dual-label approach exhibited a positive learning effect during the annotation process. Our machine learning results demonstrated that labeled images using the dual-label approach improved the accuracy of a classifier trained with the annotation results. We discuss the perceived usability and workers' confidence in challenging image annotations. The findings presented significant insights into the future development of crowdsourcing annotation tools for challenging image annotation.

### ACKNOWLEDGMENTS

This work was supported by JST CREST Grant Number JP-MJCR17A1, and JST, ACT-X Grant Number JP-MJAX21AG, Japan.

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
