# OpenReview forum: "DualLabel: Secondary Labels for Challenging Image Annotation"
_graphicsinterface.org/Graphics_Interface/2022/Conference — GI 2022_

### Official Review · Reviewer_1wgk · 2022-04-11
**Interesting system, but the consideration of psychological burden could be deeper.**

**Rating:** 6
**Confidence:** 3

**Review:**

This paper discusses DualLabel, an annotation tool that allows non-expert annotators to assign two labels to an image. This may be useful in cases when the annotator is unsure of the exact label, but is confident it is one of the two. A user study and machine learning experiment provide insight into the impacts of DualLabel when compared to single-label tools, and the results show that using multiple labels can lead to more data collection and increased accuracy of a classifier.

I think the paper is well-written and the results are thoroughly presented and discussed. I do think the related work could include more discussion of multi-labeling within a single image and ways of resolving (rather than just indicating) ambiguity to improve label quality (e.g., adjudications amongst crowdworkers [A]). There are some things that could be clarified and improved, but overall I think the paper is a nice contribution.

Overall, I love the idea of reducing the psychological burden of making a decision through DualLabels. Having a binary high/low confidence labeling system certainly seems like it would save time, especially given theories like the paralysis of choice. I think it's a great result that DualLabels led to more data in the same amount of time. However, I think the exploration of psychological burden could be much deeper, especially since it's one of the main motivations. For example, how much time did participants spend thinking over their labels before or after assigning them? One idea could be to estimate this by analyzing click times.

I also wonder why DualLabels was not compared to other techniques that require a confidence score (e.g., single-label with a confidence slider and dual label with a confidence slider)? This may be to reduce transaction costs/the psychological burden, but even having discrete options for confidence, like high/medium/low, would allow for more nuanced dual labels (like two labels that the annotator is equally confident in). This may further improve classification without too much additional time.

I am a bit skeptical of some of the results, especially those presented in Figure 6. The error bars for the Easy Dataset condition do not overlap, which would suggest a significant difference between them. The SD is definitely higher, but I'm wondering if the data was not normally distributed (which is often the case with time). In that case, the Wilcoxon signed rank test should be used instead of the paired t-test. The authors should check for normality and state whether the normality assumption was violated.

Other:
- References should be in alphabetical order (see the sample PDF online)
- Grammatical error (missing "A" at the beginning of Section 8)

# References
[A] Schaekermann et al. 2019. Understanding Expert Disagreement in Medical Data Analysis through Structured Adjudication. CSCW 2019. https://doi.org/10.1145/3359178

---

### Official Review · Reviewer_BCUt · 2022-04-11
**Incomplete analysis**

**Rating:** 4
**Confidence:** 4

**Review:**

I am of two minds about this paper. On the one hand, it suggests a reasonable interface for eliciting more labeling information from users, with a decent experiment and potentially some implications for data quality. On the other hand, its claims are more general than warranted given the findings, and the very low quality of secondary labels needs more scrutiny. I have rated it "marginally below threshold" because I think it would benefit from more analysis than can be done in a short revision cycle. This rating has large error bars and I could be persuaded that the paper could be published. I look forward to seeing the other reviews.

DualLabel proposes improving image annotation efficiency by allowing users to assign a high-confidence label plus, optionally, a low-confidence label when the labeler is not very confident in the first label.

The paper provokes some interesting questions about label accuracy and how to conduct evaluations in the presence of noisy data. I am
concerned that the specific findings are an artifact of the low accuracy of training data, such that noise (secondary labels) improves things. I weakly recommend the paper hold off on publishing until the question is investigated further. Merely indicating unreliability of data (possibly with a randomly chosen secondary label) might have similar outcomes.

The lower cognitive overhead of the dual-label task is worthwhile. I would have liked to see a comparison with an "unsure" button (i.e., the user would still provide a primary label, but instead of a secondary label, they would optionally flag a label as uncertain). It seems that this approach could have much of the benefit of dual labeling but potentially be even faster. I am dubious about the value of the highly-inaccurate secondary label.

At times, the paper appears to claim that the dual-label interface creates more data for machine learning. In one sense, this is trivially true: we had one label before, now (sometimes) we have two. The number of data points is not actually different, though. If we were to treat all labels the same (both primary and secondary labels function as ground truth) performance is likely to suffer. The observed improvement in performance should be attributed to the details of the training, which involve incorporating both labels into a single output vector. Because the primary labels were so inaccurate (in both single and dual labeling cases), introducing some uncertainty is likely to improve things.

I recommend some additional analysis that could help bolster confidence that we are seeing a real effect. Both the easy and difficult dataset had quite noisy primary labels. The secondary labels were such low quality that one might wonder whether they were much different from noise: that is, could similar results have been obtained by taking random labels instead of the selected secondary labels? The training protocol differed between images with a secondary label and images without, and perhaps the mechanism for increased performance was that the primary labels were less accurate (and indirectly indicated as such) when a secondary label was provided. I did not see a comparison of accuracy in the two-label case between annotations with one and two labels. (I do not mean something like figure 7, which compares the single and dual approaches, but rather a breakdown of the dual-approach data between cases where one label was given and where two labels were given.)

Because the secondary labels are so inaccurate, I wonder whether they are measurably different from chance. One idea would be to assign random labels to augment primary labels and see whether there is a significant difference between the accuracy of such labels and the secondary labels the subjects actually assigned. A little care would be needed with this experiment, since the secondary labels were not always assigned, and you would want to match the accuracy profile of the primary labels in this case (since your secondary label would not be the same as the primary label).

It would be possible to synthesize higher-accuracy datasets (by discarding a fraction of data points known to be inaccurate). At some point, the incorrect secondary labels would cease to provide any benefit. It could be interesting to see where the crossover occurs. This experiment is apt to be quite time-consuming, though, and may not provide any additional insight once the investigations mentioned above have been done.

Additional comments:

"A novel annotation tool": this is not a very strong contribution. What in the interface is new? Many commercial tools allow multiple labels to be assigned to photographs, though not necessarily in the context of machine learning.

"...the dual-label annotation task can collect 66.8% ... more labeled images": this is profoundly misleading. You do not have additional labeled images; you have additional low-quality labels for images you already had primary labels for. Moreover, the secondary labels are extremely low quality, around 10%-14% accurate. From a pool of 12 (11) possible labels, this is only a little above chance.

"data difficulty does not affect the amount of collected data": this is quite a strong conclusion from this small experiment. I would not assume the finding will generalize.

2.2: "wildly used" Probably the authors mean "widely".

Figure 11b: "How helpful are the labeling tasks?" As a user, I would have no reasonable way of answering this question. Helpful to whom and for what purpose? Users do answer such questions, but this seems to elicit only a measure of positive sentiment and not anything more definite than that.

You aggregated data by breed for significance testing. Any reason for that? I do not have a specific objection, but it seems as if the clustering could affect the results.

Edited to add: The HCI contribution of this paper is minimal. The most interesting finding is the reduced task completion times (and implied lower cognitive burden) in the dual-label condition. I am concerned that the dual-label ML findings have been oversold and the second label is not actually helping, and the conclusions in the paper do not actually follow from the analysis done. I am recommending rejection on that basis.

---

### Official Review · Reviewer_QF3h · 2022-04-13
**Interesting paper, some parts need more justification**

**Rating:** 7
**Confidence:** 4

**Review:**

This paper tackles an important issue for machine learning (ML) – that of improving the quality of manual image annotation. It focuses on expediting the data annotation process by exploring the relationship between data quality and labeling efficiency for challenging image annotation tasks. The major contribution of this paper is a novel dual-label image annotation tool that allows users to assign two labels to individual images (a high-confident and a secondary low-confident label). The evaluation of the dual-label image annotation approach compared to the baseline approach (single-label annotation) for easy and difficult datasets (dog breeds, in this case) shows that the secondary labeling approach is more accurate and requires less time to complete the image annotation task.

Overall, the functionality and usage of the dual label system is understandable and has been well-explained in most cases. I did not find this paper to make a ground-breaking contribution to HCI, but the incremental improvement in data labeling using dual-label image annotations is solid and is worthy of publication at GI.  The authors have contributed a dual-label approach for annotating the images for one domain (dog breed), but whether this approach is generalizable to other domains needs more clarification (for example, where users may need to have more domain-specific expertise).

I have a few suggestions below for improving the clarity of the prose and making the paper even stronger:

1.	Overall, the problem statement is mostly clear, including the motivation to devise a dual-label image approach and the need to expedite and make the annotation process usable for the users. However, in some instances, there were incomplete reasons from the related work for deriving some claims (e.g., expectation of improving the accuracy of classifier trained with more images from dual-label versus single-label). A more clear articulation of the gaps based on prior work is needed to make a stronger case for Dual-Label. Similarly, I was wondering what could have been the struggle with the partial label learning technique that dual-label has solved because partial labelling slightly aligns with the dual-label approach?

2. Some of the system design decisions need more justification - for example, the statistical definition of high-confidence and low-confidence labels is missing; did the authors consider some threshold or confidence scores that distinguish a high-confidence label from a low-confidence label? I did not see any justification in selecting the AlexNet dataset – why was this particular dataset selected?  It would also be helpful to know the authors’ decision of choosing weight values i.e., 0.5 for low-confidence scores and whether or not that was tied to some experimentation.

3.	The user study is mostly clear, but I wasn’t sure to what extent the authors recruited users who would have domain-specific knowledge about dog breed; this needs to be clearly explained. Furthermore, I am concerned that the task description given to the users seems to be giving away information with phrases such as, “if you feel more confident or less confident”, which could have biased some of the labelling.

4.	It was great to see that the dual-label approach took less time during difficult tasks. However, some the results based on some of the other measures were not significant. It would be helpful to know some of the reasons behind this.

5.	Please proofread the manuscript for grammar consistency. There are some unconventional words and phrases (e.g., “un-pretrained model”) which should be reviewed. A couple of typos appeared in Section 2.2 where Wu et al.’s model was described:, “ the validity of our model is..”- I wonder if that’s the validity of their model? Terms such as “approach to provide better data” seems unclear – what is “better data” in this context?

---

### Decision · Program_Chairs · 2022-04-17

Accept